# Landscape-scale spatial variations of pre-Columbian anthropogenic disturbances at three ring ditch sites in French Guiana

**Marc Testé**[1]*, **Julien Engel**[1], **Kevin Mabobet**[1], **Mickael Mestre**[2], **Louise Brousseau**[1]

**1** AMAP, CIRAD, CNRS, INRAE, IRD, Montpellier University, Montpellier, France, **2** French National Institute for Preventive Archaeological Research (INRAP), Matoury, France

* marc.teste@ird.fr

## Abstract

In the past two decades, repeated discoveries of numerous geometric earthworks in interfluvial regions of Amazonia have shed new light onto the territorial extent and the long-term impact of pre-Columbian populations on contemporary landscapes. In particular, the recent development of LiDAR imagery has accelerated the discovery of earthworks in densely forested hinterlands throughout the Amazon basin and the Guiana Shield. This study aimed to evaluate the extent and landscape-scale spatial variations of pre-Columbian disturbances at three ring ditch sites in the French Guiana hinterland. We carried out extensive soil surveys along approximately 1 km-long transects spanning from ring ditches through the surrounding landscapes, and drawn upon multiple indicators, including archaeological artifacts, macro- and micro-charcoals, soil colorimetry, and physicochemical properties to retrace the pre-Columbian history of these sites in terms of occupation periods, anthropogenic soil alteration, and ancient land use. Our results revealed a perennial occupation of these sites over long periods ranging from the 5th and 15th centuries CE, with local enrichments in chemical indicators (Corg, N, Mg, K, Ca) both within the enclosures of ring ditches and in the surrounding landscapes. Physicochemical properties variations were accompanied by variations in soil colorimetry, with darker soils within the enclosure of ring ditches in *terra-firme* areas. Interestingly however, soil properties did not meet all the characteristics of the so-called Amazonian Dark Earths, thus advocating a paradigm shift towards a better integration of Amazonian Brown Earths into the definition of anthropogenic soils in Amazonia. Soil disturbances were also associated to local enrichments in macro- and micro-charcoals that support *in situ* fire management that could be attributed to forest clearance and/or slash-and-burn cultivation. Taken together, our results support the idea that pre-Columbian societies made extensive use of their landscapes in the interfluvial regions of the French Guiana hinterlands.

## Introduction

Covering more than 5,500,000 km$^2$, the Amazon rainforest is one of the largest tropical forests in the world. Despite increasing attention to its extraordinary biodiversity, the Amazon

**Data Availability Statement:** This study complies with the Open Access and FAIR Principles. The protocol is available from protocols.io under DOI:

dx.doi.org/10.17504/protocols.io.14egn768pv5d/
v1. The geoarchaeology data are available from
Zenodor under DOI: dx.doi.org/10.5281/zenodo.
10430417.

**Funding:** DOPAMICS is funded by the European
Union under grant no. 101039272. Views and
opinions expressed are those of the authors only
and do not necessarily reflect those of the
European Union or the European Research Council
Executive Agency (ERCEA). Neither the European
Union nor the granting authority can be held
responsible for them. The funders had no role in
study design, data collection and analysis, decision
to publish, or preparation of the manuscript.

**Competing interests:** The authors have declared
that no competing interests exist.

rainforest remains largely unexplored and has long been considered by environmental researchers as an undisturbed, pristine forest. From another point of view, archaeological research has shed new light on pre-Columbian occupation in Amazonia, suggesting high population density and long-term impact on soil and vegetation. This recent paradigm shift is accompanied by hot debates over the territorial extent of pre-Columbian societies, their long-term impact on contemporary wild landscapes and underlying land-use strategies [1, 2].

## The territorial extent of pre-Columbian societies in Amazonia

Geometric earthworks represent one the most meaningful proxy of the territorial extent of pre-CE 1492 societies in Amazonia. However, they are often hidden below the forest cover, and, until recently, they were mostly discovered by pedestrian surveys in easily accessible areas. This long fueled the idea that pre-Columbian societies settled mainly along major rivers and coastal areas [3], which represent a very restricted fraction of the Amazon region. However, both extensive deforestation and fast development of airbone imagery technologies (e.g., LiDAR) in remote, densely forested areas has accelerated the discovery of numerous earthworks all over the region, thus arguing for a reconsideration of the pre-Columbian population density both in the Amazon floodplains and in interfluvial (*terra-firme*) forests [4–6]. On the southern rim of the Amazon, geometric earthworks are organized into complex and continuous networks [4, 7, 8]. Pärssinen et al. [9] reviewed the distribution of earthworks in the Brazilian state of Acre (western Amazonia), totalling about 200 sites over a region more than 250 km across. In 2022, Prümers et al. [5] mapped monumental settlements in a savanna-forest mosaic in Llanos des Mojos (Bolivia, western Amazonia), arguing for a region densely and continuously populated. Other authors, on the contrary, argued that human occupation was heterogeneous in western Amazonia. For example, McMichael et al. [10] surveyed 55 sites over a surface of 3,000,000 km$^2$ in central (Brazil) and western Amazonia (from Peru to Bolivia) and detected only sparse evidence of human impacts in interfluvial forests. The distribution of earthworks is not restricted to the Amazon basin (that is, the region drained by the Amazon River). In French Guiana (Guiana shield, northeast Amazonia), ring ditches are found throughout the hinterland and they are locally named "Montagne Couronnées" (abbreviated "MC"). In the 1990s, about 50 sites of ring ditches were already identified [11]. Some of them have been the subject of dedicated archaeological or paleoenvironmental research [12–15], while many others have been discovered more recently through preventive archaeology operations [16] or mapping operations in managed or protected areas.

## Geometric earthworks: Temporality and functions

Radiocarbon dates suggest that geometric earthworks were built over very long periods of time, varying between regions and sites from the first millennium to the decline of pre-Columbian societies, see S1 Table. Earlier periods of ring ditch construction dating back to BCE 1600 were detected in the Acre, Brazil [17], while occupation prior to ring ditch construction has also been suggested in some regions such as Llanos de Mojos, Bolivia [18]. Geometric earthworks are of various forms and sizes. Although their exact function remains uncertain, two kinds of architectural-function categories seem to stand out: ceremonial centers on one side and fortified villages on the other [4, 9]. Ceremonial centers, such as geoglyphs found in the Acre, are characterized by almost perfect geometric forms and symmetry combining different shapes (circles, squares, hexagones), sparse ceramic deposits, the presence of votive artifacts, and no alteration of soils [17]. On the contrary, ring ditch sites are commonly attributed to fortified (i.e., palisaded) villages based on their asymmetrical—irregularly shaped—enclosures, high-density of ceramics, and soils enriched in organic matter and burning deposits (i.e.,

charcoals), suggesting permanent habitation. Ring ditches are often attributed to Arawak speakers, but determining the exact earth builders is hard due to the linguistic and cultural diversity in Amazonia [4, 6, 19]. Although geographically distant, the ring ditches of the Guiana shield share geometric features like those found in the Amazon basin. They are called 'kalana tapele' (i.e., 'Kalana's ancient villages') in Wayãpi oral traditions, a mysterious cultural group that lived in French Guiana until the 18th century. In 1982, Grenand and Dreyfus recounted the story of a Wayãpi chief who mentions Kalana villages: *'Throughout the region, there were many Kalana. They built villages that they protected with ditches two meters wide and one meter deep. The bottom was planted with stakes. Those who did not know fell and died.'* [20]. The last traces of the Kalana (also named 'Carannes' by Europeans) are found in French national archives overseas that relate a war between Europeans and the Kalana in 1703–1704 [21, 22] who probably disappeared soon afterwards.

## Environmental impact of pre-Columbian societies and land use

The long-term impacts of pre-Columbian occupation extend beyond the mere presence of geometric earthworks. Both below and above ground features provide information on the extent of human impact and land use strategies. Above-ground, modern vegetation helps to understand how pre-Columbian societies lived in their environment in terms of subsistence and management of natural resources. For example, spatial variations in present-day vegetation that are correlated with pre-Columbian occupation are interpreted as evidence of ancient landscape management for plant cultivation [23, 24]. Such vegetation turnover is commonly interpreted as 'domesticated landscapes' [25] or 'cultural forests' [7, 26], where secondary forests replaced primary forests. In 1989, W. Balée [26] has estimated that more than 11% of Brazilian *terra-firme* forests were anthropogenic. In the same vein, some useful plant taxa and crop species are significantly more abundant in forest areas containing evidence of pre-Columbian occupation, such as ring ditch sites, compared to supposedly natural forest areas without evidence of human occupation [27–29]. These observations are also supported by palynological records, which reveal the presence of major crops (e.g., maize, squash, manioc) during periods of human occupation of ring ditch sites [18]. Based on the abundance of charcoal, paleoenvironmental research has documented repeated fire activity in Amazonia through the Holocene. Because natural fires are rare in Amazonia, it is commonly admitted that the presence of charcoal in palaeoecological records can be attributed to anthropogenic burning [30], although fires in evergreen forests probably result from a close interaction between human activity and natural climate variations (seasonality and El-Ninō climate oscillation); drier periods being more favorable to burning [31]. However, whether burning activity was associated with large-scale deforestation is unclear and should be considered in light of natural, climate-driven vegetation dynamics. In the ring ditch region of Llamos de Mojos (Bolivia), pre-Columbian earth builders may have first settled in preexisting savannas they maintained open through slash-and-burn during late-Holocene forest expansion [32]. The decline of pre-Columbian populations was characterized by a decrease in burning activity ~600–500 B.P. (i.e., CE 1350–1450) and associated with forest expansion [18]. A similar hypothesis has been proposed in the Acre, where the earthworks were probably built in bamboo forests. Only localized forest clearances were carried out without involving substantial deforestation [33]. Soil properties may also provide interesting information on the extent of anthropogenic disturbances in contemporary landscapes. In Brazil, pre-Columbian settlements are commonly associated with the so-called Amazonian Dark Earths, ADEs (i.e., '*terra preta*'), which are acknowledged as highly fertile soils. Patches of ADEs are frequent along the Amazon River and its tributaries [34] but they are rare, or still undiscovered in more distant regions and

hinterlands [1]. Model-based predictions suggest that only 3.2% of Amazonian forests might be propice for *terra preta* formation [35]. They further confirm that the distance to rivers is an important predictor, the occurrence of *terra preta* being more probable within 10 km of a river. ADEs soils are often considered as anthropogenic soils formed by the addition of inorganic and organic debris (e.g., bones, biomass wastes, excrements) and burning activity [36, 37]. Indeed, ADEs are commonly enriched in organic matter and nutrients [37, 38]. Even if the anthropogenic origin of ADEs is still debated [37–40], it is commonly admitted that long-term landscape management alters soil characteristics to varying degrees, ranging from soils with slight to moderate anthropogenic modifications (i.e., 'terra mulata', [41–43]) to anthropogenic soils or 'anthrosols' containing strong anthropogenic modifications (i.e., 'terra preta').

However, soil properties in interfluvial pre-Columbian sites less propitious to *terra preta* formation have been less documented. Furthermore, the specific location of French Guiana in northeast Amazonia also questions the extent of soil alterations at pre-Columbian sites, and empirical data are profoundly lacking.

The ring ditches of the Guiana shield have been poorly studied compared to those of the Amazon basin, thus constituting a considerable knowledge gap. Whether ring ditches located in the Guiana shield are associated with anthropogenic soil alterations and/or burning activities remains poorly known. Furthermore, the extent of spatial variations in anthropogenic disturbances within forest landscapes surrounding ring ditches has rarely been assessed. This knowledge may have important implications in terms of ecosystem resilience and sustainable land use.

In this study, we investigated the landscape-scale extent and spatial variations of pre-Columbian anthropogenic disturbances at three ring ditch sites in French Guiana. We carried out a continuous, standardized soil sampling along approximately 1 km-long transects that extend across ring ditch enclosures and surrounding forest landscapes. We built upon an integrated methodology that combine soil and charcoal analyses to address the following questions:

- What were the occupation periods of the sites? Do these periods coincide with previously documented periods of ring ditch construction in Amazonia?

- Do contemporary forest landscapes surrounding ring ditches contain evidence of long-term pre-Columbian disturbances? How does the spatial distribution of anthropogenic disturbances vary in forest landscapes surrounding ring ditches?

- Do soil properties meet the characteristics of Amazonian Dark Earths?

- Do soil properties inform ancient land use and landscape management?

## Materials & methods

### Study area and ring ditch sites

French Guiana lies to the east of the Guiana shield (northeast Amazonia), a 1.7-billion-year-old Precambrian craton characterized by a soft relief alternating between low hills and slopes on ferralsol (i.e., *terra-firme*) and flooded lowlands (i.e., bottomlands) on hydromorphic gleysols. French Guiana is located in the Guianan Lowland Moist Forests ecoregion (bioregion NT21; ecoregion 465 according to One Earth: https://www.oneearth.org/ecoregions/guianan-lowland-moist-forests/) and is covered by approximately 97% of the evergreen Amazon rainforest. This highly diverse rainforest grows in the Af climate according to the Köppen—Geiger classification [44, 45], which is marked by an annual rainfall ranging between 2000 and 4000

mm/year, a wet season that extends from December to July (interspersed with a 'little summer' in March) and an average annual temperature of 26˚C. This study focused on three ring ditches in the French Guiana hinterland: MC87, Nouragues (NOUR), and Mont Galbao (GALB). These sites are distributed along rainfall and altitude gradients ranging from ~2700 to ~3700 mm/yr (Fig 1).

**MC87.** MC87 is located in the Kourouaï River watershed (4.0572038˚N; -52.0863977˚E, WGS84), Fig 1. The site occupies the top of one of many palaeoproterozoic bedrock hills of the zone (alt: 85 m AMSL) and is surrounded by waterways and bottomlands [14]. The local rainfall at this site approaches 3800 mm/year. A previous archaeological and micromorphological survey has been carried out on this site, providing background information [14]. The 2 m deep by 2–3 m wide ditch is located on the top of a hill and encircles an area of almost 1 ha. Radiocarbon analyses dated the pre-Columbian occupation between CE 500 and CE 1100. No human occupation has been reported at this site during the colonial period. Despite a forest exploitation road to the east of the ring ditch, the site has been spared contemporary logging operations and had been impacted by economic activities only at the eastern end of the transect (i.e., the last pit 'C20' in Fig 1).

**Nouragues (NOUR).** The Nouragues site lies on the flattened summit of a precambrian granite plateau in the watershed of the Arataï River (4.0790943˚N; -52.6713032˚E, WGS84), Fig 1. The climate at this site is slightly less humid than at MC87, with 3200–3400 mm annual rainfall. The site is located in the CNRS Research Station within the Nouragues Natural Reserve, a fully protected area since 1995. This station has been the subject of numerous ecological and archaeological research projects, including the LONGTIME project [46]. This ditch studied here is located in the zone named 'grand plateau' and is made up of a shallow, partially closed ditch 0.5 to 1 m deep and 2 m wide (Fig 1). Today, only an arc of a circle curved over around one hundred meters is still visible through LiDAR. Radiocarbon dating of the ditched site indicate occupation between CE 650 and CE 1550 [15]. An Amerindian presence in the region was noted until the end of the 18th century: the 'Nouragues' or 'Norak' Tupì speakers [20].

**Mont Galbao (GALB).** The Mont Galbao site is located on a flat part of the eponymous mountain range, within the metamorphic chain Inipi-Camopi (3.5958754˚N; -53.2924755˚E, WGS84), Fig 1. This area lies upstream of the Makwali River watershed and is part of the protected area of the French National Park (PAG). In 2020, a dozen ring ditches were discovered through a LiDAR mapping operation ordered by the PAG. The studied ring ditch is an ellipsoidal and extends more than 300 m long and 130 wide. It is up to 3 m deep with a flattened outer edge in its southern section. Due to the remoteness of this site, which is not deserved by public transports and is only accessible by helicopter, no archaeological excavations or radiocarbon dating have been carried out on the site before the present study.

## Sampling design and soil sampling

This study complies with local and European regulations. Archaeological surveys were authorized by prefectoral decision n˚2022–73 of 12 September 2022 (Principal Investigator: Louise Brousseau; Operations Manager: Marc Testé). The access to the French National Park 'Parc Amazonien de Guyane' for scientific purposes was authorized by authorizations n˚1187–22 of 23 September 2022 and n˚2023–003 of 13 February 2023. In accordance with the Commission Delegated Regulation (EU) 2019/829 of 14 March 2019, the import of non-European soil samples into the European Union territory for physicochemical analyses was authorized by an Official Authorization Letter of 27 July 2023. Additional information regarding the ethical,

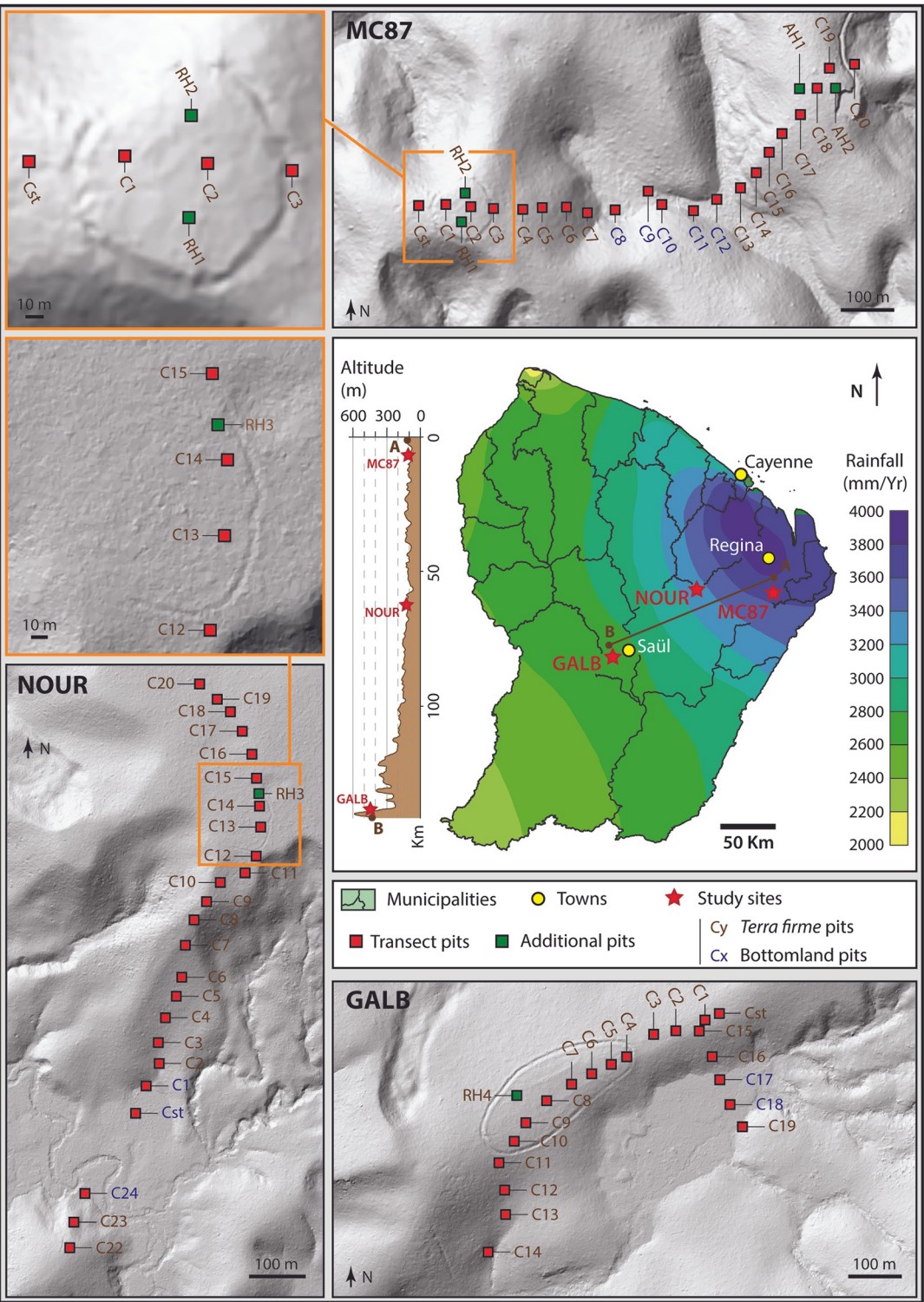

**Fig 1. Location of study sites and sampling design.** The LiDAR Digital Terrain Model (DTM) of MC87 was provided by the National Forest Office (Office National des Forêts, ONF). The LiDAR DTM of NOUR was provided by LONGTIME project. The LiDAR DTM of GALB was provided by the Parc Amazonien de Guyane (PAG).

cultural, and scientific considerations specific to inclusivity in global research is included in the S2 File.

At each study site, we performed a continuous and standardized soil sampling along approximately 1 km-long transects that extended through the ring ditch hill (RH) and the surrounding forest landscape with little or no evidence of pre-Columbian disturbance (i.e., no earthwork, absence of or few shards and charcoals in treefall mounds) (Fig 1). For each transect, the soils were sampled every 50 meters with an auger (pits named: Cstart, C1, C2, etc.). Additional pits were also sampled within the ditched enclosure (pits named: RH1, RH2, RH3, RH4) and on an adjacent hill (pits named: AH1, AH2). For each pit, 5 cm-thick soil samples were collected from the surface (0-5cm) to a depth of 120 cm (115-120cm), i.e., a total of 24 samples per pit. However, given the absence of archaeological artifacts in deep samples, we focused on the first 60 centimeters in the following analyses.

## Soil analyses

**Archaeological artifacts, soots and macrocharcoals.** The bulk soil samples were dried in an oven (50˚C), weighed and sieved (<5 mm) to remove roots and stones and extract archaeological artifacts (i.e., ceramic shards) and macroscopic charcoal (> 5mm). Macroscopic charcoals (~2–5 mm) and soots were extracted by hand from the sieved fraction. Given the reduced volume of dry soil obtained every 5 cm, the presence of archaeological artifacts, soots, and macrocharcoals was evaluated as discrete variables (i.e., presence/absence, encoded '1/0').

**Colorimetry.** The color of the soil samples was measured using a PANTONE CAPSURE colorimeter implemented with the Munsell color chart. Each sample was subjected to five consecutive measures to determine the dominant color. The samples were then separated into two different fractions: a 100–200 g fraction was reserved for physico-chemical analysis. The remaining fraction was kept for evaluation of microcharcoal abundance.

**Microcharcoal abundance.** The abundance of microcharcoals (160 μm—2 mm) was evaluated every 5 cm from 0 to 60 cm on 8 to 9 pits in *terra-firme* areas at each study site, for a total of 300 microcharcoal sample counts. To deflocculate the clays, 2.5 $cm^3$ of soil was exposed to Sodium hexametaphosphate at 40 g/L and incubated during 12 hours with an orbital shaker. The samples were then sieved (<160 μm) and dried in an oven at 50˚C for 12 hours [47]. Microcharcoals were counted using a Zeiss stereo microscope at 40x magnification.

We tested whether microcharcoal varied between sites (MC87, NOUR, GALB), landscape-scale localization (i.e., ring ditch enclosure and ditch, ring ditch hilltops and slopes, adjacent hills) and depth through a Generalized Linear Model (GLM) using R version 3.5.1. GLM was encoded using the functions 'glm()' and 'drop1()', with 'Site', 'Localization' and 'Depth' as explanatory variables, and 'Microcharcoal count' as response variable (Microcharcoal count ~ Site + Localization + Depth).

**Physico-chemistry.** For each pit, physicochemical analyses were carried out on one sample every 10 cm between 5 cm and 60 cm depth in *terra-firme*, totaling 6 samples per pit: 5–10 cm; 15–20 cm; 25–30 cm; 35–40 cm; 45–50 cm; 55–60 cm. In the bottomlands, dry soil samples were always well below 100 g due to important waterlogging. To remedy this, we pooled surface samples up to 20 cm and then every 10 cm up to 60 cm depth, thus totaling 5 samples per pit: 0–20 cm; 20–30 cm; 30–40 cm; 40–50 cm; 50–60 cm.

A total of 410 physicochemical samples were sent for analysis to the SADEF laboratory (Aspach-Le-Bas, France), which is accredited for the import of non-European soils. A total of 12 soil properties were evaluated, including granulometry (i.e., particle size distribution) and chemical properties (pH, major and minor elements). Table 1 synthesizes the methods used

**Table 1. Soil properties, abbreviations and methods.**

| Property | Abbreviation | Method | Standard |
|---|---|---|---|
| Granulometry (i.e., particle-size distribution) | PSD | Pipette method without decarbonation | NFX 31–107 |
| pH | pH | $H_2O$ extraction ($pH_{H20}$) | NF EN ISO 10390 |
| Total Carbon | Ctot | Dry combustion method | NF ISO 10694 |
| Organic carbon | Corg | | |
| Total carbonates (i.e. carbonates, mainly $CaCO_3$) | Cinorg | Volumetric method | NF EN ISO 10693 |
| Total Nitrogen | Ntot | Dumas method | NF ISO 13878 |
| Phosphorus ($P_2O_5$) | P | Olsen method | NF ISO 11263 |
| Potassium ($K_2O$), Magnesium (MgO) | K | Inductively coupled plasma-atomic emission spectrometry (ICP-AES) | NFX 31–108 |
| Calcium (CaO) | Mg | | |
| Sodium (exchangeable $Na_2O$) | Ca | | |
| | Na | | |
| Aluminum (exchangeable) | Al | Molar KCl extraction and ICP-AES | NA |

and standards, when applicable. The amount of soil carbonates (Cinorg) was null in all soil samples and was excluded from our analyses.

We tested whether soil properties varied between soil types (ferralsols, hydromorphic gleysols) through Generalized Linear Models (GLM) using R version 3.5.1 (Model 1: Soil property ~ Soil type). We also tested whether the physicochemical properties of the *terra-firme* soils varied between study sites (MC87, NOUR, GALB), landscape-scale localization (i.e. ring ditch enclosure and ditch, hilltops and slopes of the ring ditch, adjacent hills) and depth (Model 2: Soil property ~ Site + Localization + Depth). The GLMs were encoded using the functions 'glm()' and 'drop1()', with the different soil properties as response variables. The GLMs were completed by principal component analyses (PCAs) with packages 'ade4' and 'FactoMineR' to visualize variations in soil properties across soil types, study sites and localization in a multidimensional space with *Nproperty* dimensions.

**Composite index of anthropogenic disturbances.** To quantify the extent of landscape-scale anthropogenic disturbances, we developed a composite index based on the acquired data. This index was designed to integrate information provided by the presence of artifacts (ceramics, macrocharcoals, soots), soil physicochemical properties and soil color in a single, unidimensional, metric (see S1 File).

## Radiocarbon dating

At each study site, 10 macroscopic charcoals (> 10 mg) were selected for radiocarbon dating, totaling 30 macrocharcoals. Macrocharcoals were selected from pits within or close to the ring ditch enclosure, at depths between 0 and 45 centimeters. Radiocarbon dating was performed by the CIRAM Radiocarbon Laboratory (Martillac, France). The calibration was performed using the OxCal 4.4 program [48] with the IntCal20 calibration curve [49].

## Results

### Radiocarbon dating

Radiocarbon dates obtained from the three study sites span from the mid-Holocene to the end of the pre-Columbian period; see Fig 2 and S2 Table. At MC87, five calibrated dates obtained from the pits RH2 (30–35), C4 (5–10, 10–15) and C5 (10–15, 35–40) were spread over a period ranging from 662 to 945 CE. Four dates, obtained from the pits RH1 (0–5, 10–15) and C2 (15–

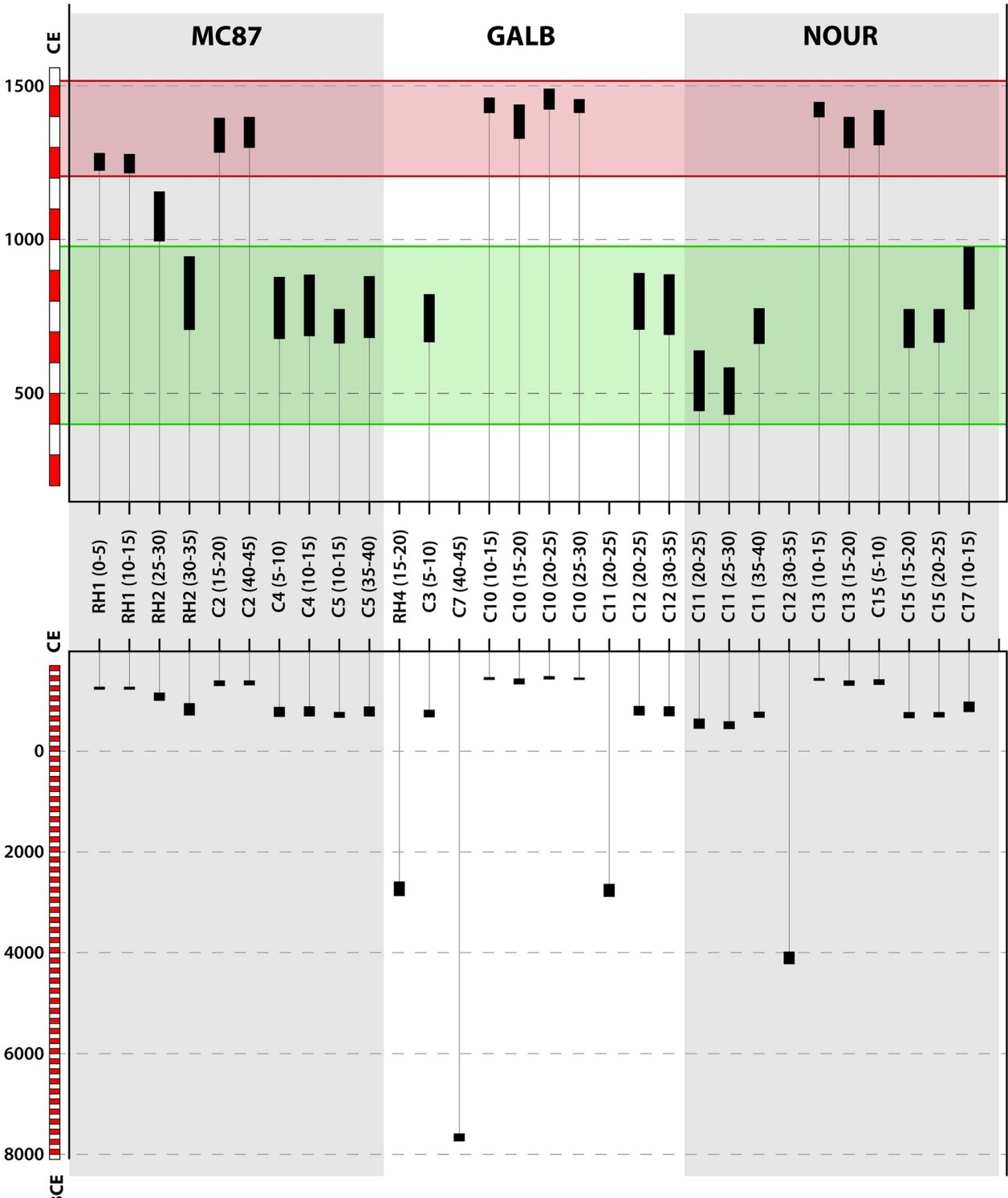

**Fig 2. Radiocarbon date ranges obtained from the three study sites at 95.4% probability.** Lower pane: all dates, upper pane: dates posterior to 430 CE. Radiocarbon data are provided in S2 Table. The green and red rectangles represent the two main time intervals in which 95% of our radiocarbon dates are grouped (excluding oldest dates BCE displayed in the lower pane only).

20, 40–45), were spread over a period ranging from 1214 to 1398 CE. One intermediate date from RH2 (25–30) was dated between 993 and 1155 CE. At NOUR and GALB, radiocarbon dates show three and four different (non-overlapping) chronological periods. At NOUR, one oldest date was obtained from the pit C12 (30–35) and dated between 4230 and 3982 BCE. The nine other dates were posterior to 430 CE, with two non-overlapping sub-periods: charcoals from C11 (20–25, 25–30, 35–40), C15 (15–20, 20–25) and C17 (10–15) spanned over a period from 430 CE to 976 CE, while dates obtained from C13 (10–15, 15–20) and C15 (5–10) spanned over a period from 1296 to 1446 CE At GALB, radiocarbon dating revealed a more complex chronology. One ancient date calibrated at 7739–7588 BCE was obtained from C7 (40–45). Two other contemporary ancient dates, ranging from 2894 to 2585 BCE, were obtained from RH4 (15–20) and C11 (20–25). The seven other dates were post-600 CE. As at NOUR, two non-overlapping subperiods emerged: three dates from pits C3 (5–10) and C12 (20–25, 30–35) ranged between 666 and 891 CE, while the four dates from C10 (10–15, 15–20, 20–25, 25–30) cover the period 1326–1490 CE.

## Archaeological artifacts, soots and macrocharcoals

Soil sampling revealed the presence of ceramic shards, macroscopic charcoals, and soots, Fig 3. At the three study sites, the presence of artifacts was mostly concentrated in *terra-firme* areas, while none or few artifacts were found in the bottomlands.

At MC87, ceramic shards were found in 8 samples from pits within the enclosure of the ring ditch (pits C1, RH1, C2) or close to the ditch (pit C3). In these pits, the proportion of samples containing ceramic shards varied between 8.3% and 36.3%, S1 Fig. At NOUR, no ceramic shards were found during soil sampling. At GALB, three samples from the ring ditch enclosure (pits C9 and C6) contained ceramic shards. In addition, an inspection of soil retained within the roots of fallen trees revealed the presence of ceramics within the ring ditch enclosures at the three sites.

In the three sites, macrocharcoals and soots were found in most of the *terra-firme* pits, both in the ring ditch and in adjacent hills. However, the highest proportions of samples containing charcoals and/or soots were found in pits from the ring ditches, S1 Fig.

At MC87, a total of 48 soil samples from 12 pits contained macrocharcoals and 27 samples from 13 pits contained soots (out of 291 soil samples from 25 pits). The highest proportions of samples containing macrocharcoals and soots were found in the pits RH2 (66.67% of the samples in the pit) and C2 (50% of the samples in the pit), respectively, which were both within the ring ditch enclosure. Secondary peaks were also detected in pits C5 (58.3% of the samples containing macrocharcoals) and C6 (41.67% of the samples containing soots), which were located on the slope of the ring ditch hill. Macrocharcoal peaks were also identified on the slope of the adjacent hill at C14 and C15 (41.67% of the samples containing macrocharcoals), while the proportion of soots was equal or below 25% (C16). The total proportion of samples containing macrocharcoals was 28.8% in the ring ditch enclosure and ditch (out of 59 samples; pits C1 to C3, RH1 and RH2), 25.0% in the ring ditch hilltop and slope outside the enclosure (out of 60 samples; pits Cstart, C4 to C7), and 13.3% in the adjacent hill (out of 120 samples; pits C13 to C20, AH1 and AH2). The proportion of samples containing soots was 17.0% within the enclosure of the ring ditch, 11.7% on the hilltop and slope of the ring ditch outside the enclosure, and 8.3% in the pits of the adjacent hill. Several pits contained neither macrocharcoals nor soots: C7 (bottom of the slope of the ring ditch), C8 to C12 (bottomland), AH1, AH2 and C19 (adjacent hilltop).

At NOUR, a total of 66 soil samples from 21 pits contained macrocharcoals and 40 samples from 16 pits contained soots (out of 296 soil samples from 25 pits). The highest proportions of

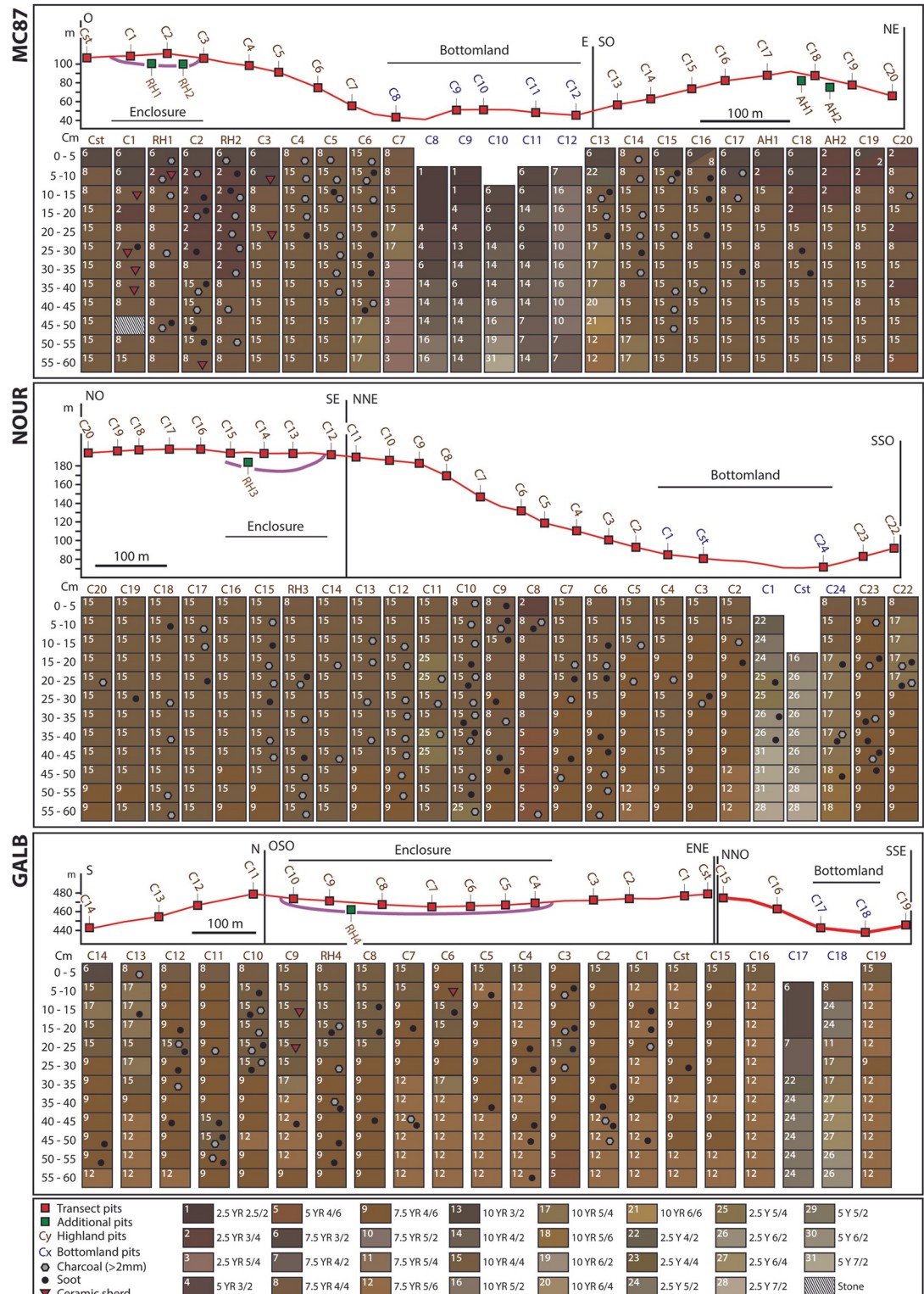

**Fig 3. Spatial distribution of soil colors and archaeological artifacts along the transects in the three study sites.**

samples containing macrocharcoals were found in pits C10 and C12 (75% and 66,7% of samples in each pit, respectively), while the highest proportion of samples containing soots was found in pit C9 (50% of samples in the pit). These three pits were located on the ring ditch: C12 was located at the outer edge of the ditch, and C10 and C9 were located on the plateau and slope of the ring ditch, outside the enclosure. Secondary peaks of macrocharcoals (33.3%) and soots (41.6%) were observed in pit C23 that is located on the adjacent hill. The total proportion of samples containing macrocharcoals was 40.0% in the pits of the enclosure and ditch (out of 60 samples; pits C12 to C15 and RH3), 19.4% in the pits of the plateau and slope outside the ditch (out of 180 samples; pits C16 to C20 on the north face of the plateau, and C2 to C11 on the south face of the plateau and slope), 3.1% in the pits of the bottomland (out of 32 samples; pits C1, Cstart and C24), and 25.0% in the pits of the adjacent hill (out of 24 samples; pits C22 and C23). Surprisingly, the total proportion of samples containing soots was higher in the pits of the adjacent hill (29.2% of samples) than in the pits of the bottomland (18.8%), the plateau and slopes of the ring ditch (12.8%) and the enclosure and ditch (6.7%).

At GALB, 20 soil samples from 9 pits contained macrocharcoals and 42 samples from 16 pits contained soots (out of 248 soil samples from 21 pits). The highest proportions of samples containing macrocharcoals were found in C10 (33.3%), RH4 (25%), C11 (25%) and C3 (25%). C10 and RH4 were located within the ring ditch enclosure, while C11 and C3 were located on the slopes and top of the ring ditch. The highest proportions of samples containing soots were found in pit C4 (41.7%), C10 (33.3%), RH4 (33.3%) and C12 (33.3%). Pits C4, C10 and RH4 were located within the enclosure, while C12 was located on the ring ditch slope. The global proportion of samples containing macrocharcoals was 8.33% in the pits of the ring ditch enclosure (out of 96 samples; pits C4 to C10 and RH4), 10.0% in the pits of the top and slopes of the ring ditch (out of 120 samples; pits C11 to C14, Cstart, C1 to C3 and C15 to C16), and 0% in the bottomland and adjacent slope (out of 20 and 12 samples; pits C17 and C18 in the bottomland and C19 on the adjacent slope). The total proportion of samples containing soots was 22.9% in the enclosure pits, 16.7% in the pits of the ring ditch plateau and slopes, and 0% in the bottomland and adjacent hill.

## Soil colorimetry

Soil colorimetry showed different colorimetric ranges between bottomlands and *terra-firme* areas, while soils from ring ditches and adjacent hills have similar color ranges. However, slight color variations were detected in the depth of soil profiles from the ring ditches to the adjacent hills.

In bottomlands, gleysol colors varied from dark brown (e.g., 2.5 YR 2.5/2, 5 YR 3/2, 7.5 YR 3/2, 2.5Y 4/2) on the surface to light gray (e.g., 2.5 Y 7/2, 5 Y 7/2) on the depth. At NOUR, one pit (C24) located in a seasonally waterlogged area was dominated by yellowish brown (10 YR 5/6, 10 YR 5/4, 10 YR 4/4).

In *terra-firme* areas, soil colors ranged from strong brown (e.g., 7.5 YR 4/6; 7.5 YR 5/6) to yellowish brown (e.g., 10 YR 4/4; 10 YR 5/4). At MC87, the dominant colors were brown (7.5 YR 4/4) and dark yellowish brown (10 YR 4/4). Dark-colored soils were found on the surface of the hilltops as revealed by the colors reddish brown (2.5 YR 3/4) and dark brown (7.5 YR 3/2), from the Cstart to C3 pits in the ring ditch (6 pits) and C13 to C19 on the adjacent hill (8 pits). However, these dark soils were thicker in the ring ditch where two pits showed deep dark soils that extended from the surface to 30 and 35 cm (pits C2 and RH2), while the thickness of dark soils did not exceed 20 cm in depth (pit C18) in the adjacent hill. The soil colors in pit 20 showed discontinuous dark soils at depths of 0–5 cm, 20–25 cm and 35–40 cm in depth, probably due to a soil disturbance caused by recent forest logging. At NOUR, the dominant color

was dark yellowish brown (10 YR 4/4) on the plateau of the ring (pits C20 to C10), while the soil profiles became gradually dominated by brown (7.5 YR 4/4) at the top of the slope (pits C8 and C9) and strong brown (7.5 YR 4/6) on the slope (from pit C9 to C2) and on the adjacent hill (C22 and C23). At GALB, the soil profiles were dominated by strong brown (7.5 YR 4/6, 7.5YR 5/6). The surface soils of the west face of the ring ditch enclosure were dominated by dark yellowish brown (10 YR 4/4) up to 30 cm deep (pits C10 to C7 and RH4). This color was also found on the surface horizon (0–5 cm deep) of the eastern face of the enclosure (pits C4 and C5) and the top of the ring ditch (C2, C1, Cstart, C15, C16).

## Microcharcoal counts

Microcharcoal counts varied significantly between study sites (p-value $< 2.2 \times 10^{-16}$), localizations within the landscape (p-value $= 4.6 \times 10^{-9}$), and depths (p-value $= 2.5 \times 10^{-12}$), S2 Fig. As expected, the microcharcoal concentration was lower in deep samples between 35 and 60 cm than in upper samples between the surface and 30 cm, with a transition at 30–35 cm deep. MC87 had the highest overall microcharcoal concentration and GALB the lowest microcharcoal concentration. Site effect remained significant when considering microcharcoal counts of the ring ditches only, i.e. after removal of the adjacent hills and testing 'Site' and 'Depth' effects only: *Microcharcoal count ~ Site + Depth* (p-value $= 3.47 \times 10^{-11}$). At the landscape scale, microcharcoal concentration was significantly higher within the ring ditch enclosures than on the ring ditch plateaus and slopes and on adjacent hills.

At MC87, one pit located within the enclosure of the ring ditch (pit C2) showed particularly high microcharcoal concentrations, whose counts varied between 655 and 1230 in the first 30 cm (Fig 4). Microcharcoal counts in this pit were above 250 in the entire soil profile, with a peak at 5–10 cm deep. In the other pits, the microcharcoal abundance was almost always below 500 and decreased in soil profiles to less than 100 at depths between 25–30 cm (pit C17) and 45–50 cm (pits RH1, C15, C19).

At NOUR, the spatial distribution of microcharcoal counts was consistent with the landscape-scale position of pits, i.e., highest peaks of microcharcoals were found within (C15, C13) or close to (C10, C7) the ring ditch enclosure (Fig 4). Four pits located within the enclosure (pits C15 and C13), on the plateau (C10) and slope (C7) of the ring ditch showed strong variations within soil profiles. Microcharcoal counts in these pits varied between 127 (C10) and 323 (C15) from the surface to 30 cm, and between 29 (C15) and 354 (C10) below 30 cm deep. Pits C13 and C15 had a peak at 15–20 cm (abundance = 303 and 323), C7 had a peak at 20–25 cm (abundance = 305) and C10 at 30–35 cm (abundance = 354). Other pits (C20, C17, C4 and C23) show lower microcharcoal abundance (almost always below 200) and flatter curves. Only the most extreme pit of the adjacent hill (C22) exhibited an intermediate pattern of microcharcoal concentration, with two peaks at 15–20 cm and 20–25 cm (abundance = 211 and 213, respectively) and microcharcoal counts below 200 in other samples.

At GALB, pits located on the hilltop of the ring ditch (C9, C7, C5, C1) showed microcharcoal counts between 59 (C7) and 217 (C7) from the surface up to 30 cm deep and between 19 (C7) and 207 (C5) below 30 cm deep, with higher concentrations between 15–20 cm (C7, count = 217) and 35–40 cm (C5, count = 207). These pits also showed a plateau of 150 to 200 microcharcoals over 30 to 40 centimeters from the surface. Except for C13, the pits located on the slope of the ring ditch (C11 and C16) and on the adjacent slope (C19) had microcharcoal counts below 150 with only slight variations in soil profiles. Only pit C13 had an irregular profile, with two peaks at 10–15 cm (count = 239) and 25–30 cm (count = 191), while samples below 30 cm deep had low microcharcoal concentrations (counts $< 60$).

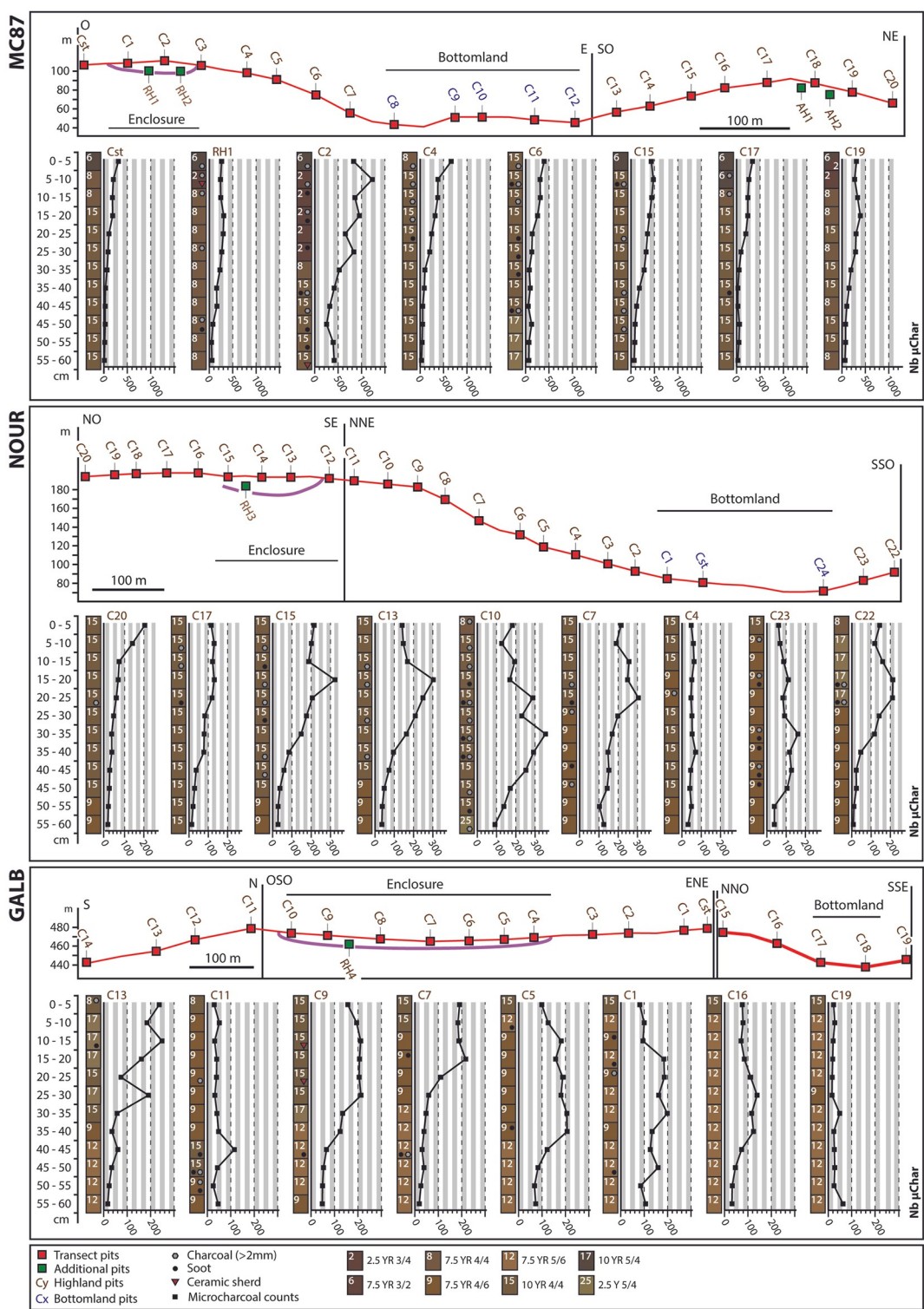

**Fig 4. Microcharcoal abundance evaluated from soil volumes of 2.5 cm³.**

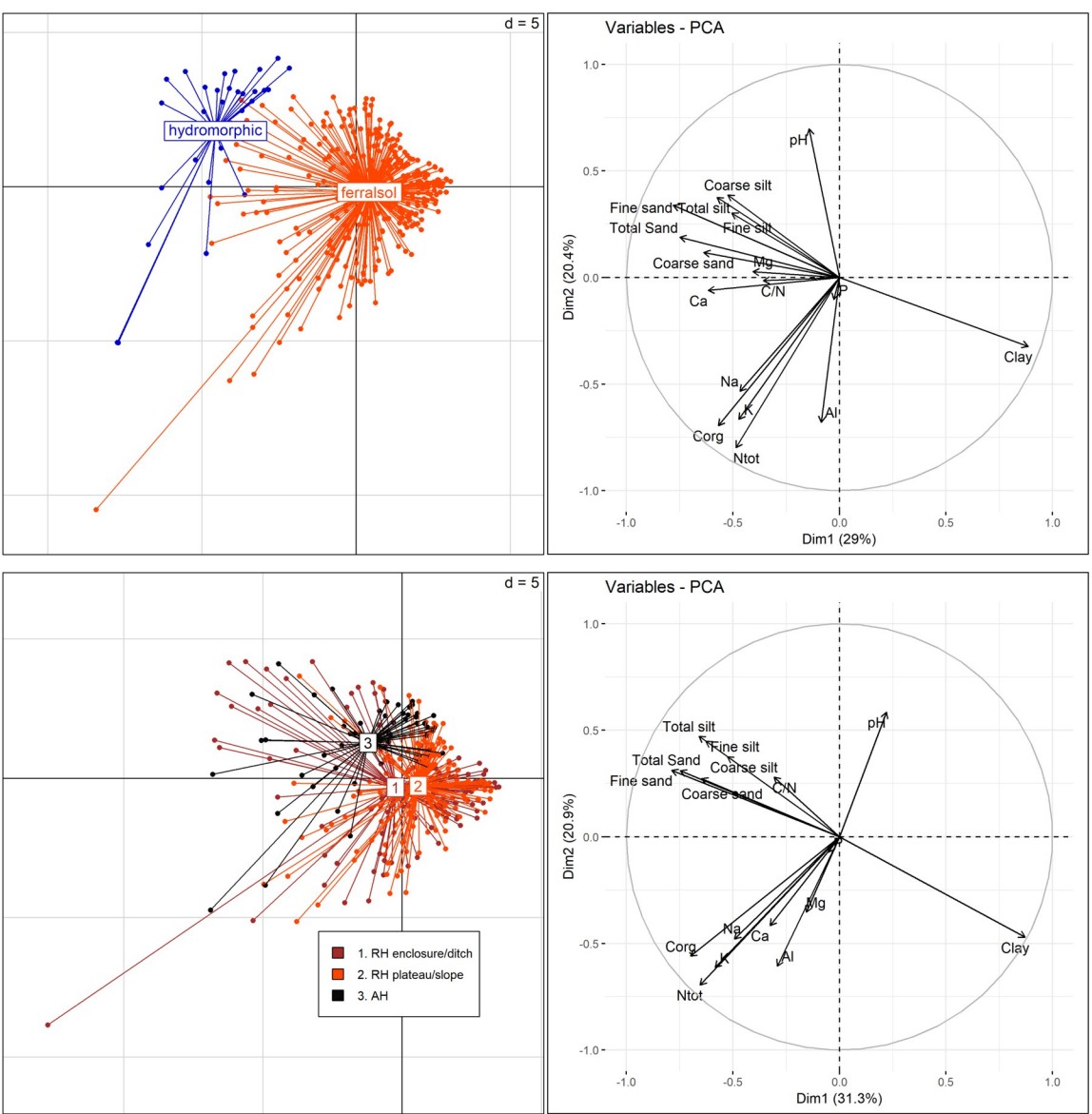

**Fig 5.** Principal Component Analyses (PCAs) illustrating variations in soil properties between soil types (upper pane) and between landscape-scale localizations in *terra-firme* areas (lower pane). For GALB, the dry mass of surface samples (above 30cm) were unfortunately insufficient for particle size analysis. Grouping by study site is provided in S3 Fig.

## Physico-chemistry

Soil properties differed between hydromorphic gleysols and ferralsols (Fig 5, upper pane). Granulometry and eight chemical properties varied significantly between soil types at a 1% threshold, S3 Table. Ferralsols contained significantly more clays while hydromorphic soils contained significantly more coarse silt, total silt, and sand. Only fine silt was not significant between soil types. Hydromorphic soils contained significantly more nutrients (Corg, Ntot, K, Mg, Ca, Na) and had a higher pH. Only P and Al did not vary significantly between soil types.

In *terra-firme*, soil properties varied significantly between study sites, landscape-scale localization, and depth range. All properties varied significantly across depth ranges at a 1%

threshold, except coarse silt, which showed significant variations at 5% threshold, and P, which did not vary significantly. Significant differences between study sites were detected for clay, sand, Corg, Mg and Al at a 1% threshold, and for P at a 5% threshold. Soil properties also varied significantly between the landscape-scale position of the pits (Fig 5, lower pane, S3 Table and S5 Fig). Pits located on the hills of ring ditches (within and outside the enclosure) contained significantly more clays, Ntot, Mg and K than adjacent hills at a 1% threshold. The enclosures of the ring ditches also contained significantly more Corg and Ca than the ring ditch slopes and the adjacent hill at a 1% threshold. Variations in Al and pH were poorly significant. Al was slightly higher in the slopes of the ring ditches than in the enclosures or adjacent hills, while pH was slightly higher in adjacent hills at a 5% threshold.

### Extent of anthropogenic disturbances

The composite index quantified the extent of anthropogenic disturbances through a multiple proxy approach that integrated artifacts (ceramic shards, macrocharcoals, soots), chemical indicators, and soil color. Variations in anthropogenic disturbances within study sites tend to increase from adjacent hills to the enclosure of ring ditches (Fig 6; S6–S8 Figs). It is also important to note that this index was less variable between 30 and 60 cm depth than between 0 and 30 cm depth where variations across landscape-scale localization and pits were more pronounced. At GALB and MC87, anthropogenic disturbances were higher within the ring ditch enclosures than outside the enclosures and adjacent hills. At NOUR, variations in anthropogenic disturbances were less pronounced, and the composite index tends to gradually decrease from the enclosure of the ring ditch to the adjacent hill.

## Discussion

This study provides an extensive, multi-proxy overview of anthropogenic disturbances in forest landscapes surrounding ring ditch sites in French Guiana. By exploring multiple indicators, including archaeological artifacts, soil color, and physicochemical properties, our study provides one of the first landscape-scale spatialization of long-term anthropogenic disturbances, not only within the enclosure of ring ditches, but in the extended surrounding landscape.

### Occupation periods

Our different cohorts of radiocarbon dates suggest two main periods of occupation that were coherent between study sites: one period extending from the 5[th] to the 10[th] century (430 CE—976 CE) and one from the 13[th] to the 15[th] century (1214 CE—1490 CE). Although we did not carry out a taxonomic identification of the macrocharcoals selected for radiocarbon dating, we dated a cohort of ten charcoals per site from different pits and depth to get a large range of dates and avoid dating the same plant. The consistency of dates obtained between the three study sites made us confident about the periods of occupation. These periods are consistent with previous radiocarbon dating reported at these sites. For example, Brancier et al. [14] reported occupation periods between 500 and 1100 CE at MC87, while the dates obtained in our study at this site ranged between 662 and 1398 CE. At Nouragues, Bodin et al. [15] reported radiocarbon dates between 688 and 1630 CE from the ring ditch of the 'grand plateau' (named 'site 9' in Bodin et al. suppl. file), while the main periods of occupation at this site extended between 430 and 1446 CE in the present study. In two sites (GALB and NOUR), occupation periods were discontinuous, echoing the bimodal distribution of radiocarbon dates reported by Bodin et al. [15] at Nouragues and by M. Mestre [50] at another ring ditch site located along the Oyapock river in eastern French Guiana. This suggests a transitional abandonment of the sites followed a population re-expansion or reestablishment several

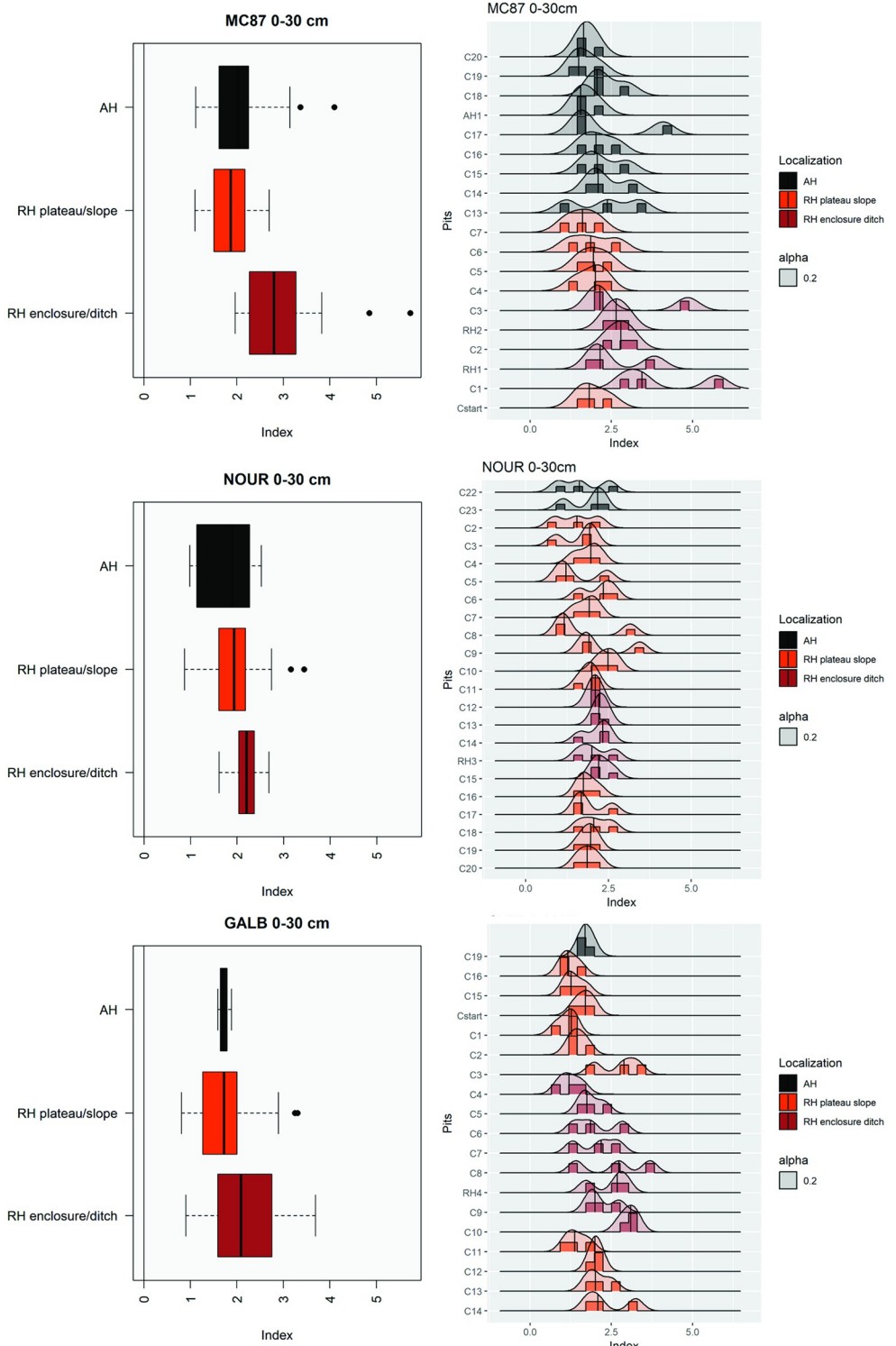

**Fig 6. Composite index of anthropogenic disturbances at MC87, NOUR and GALB between 0 and 30 cm deep.**

centuries later. The radiocarbon dates obtained in this study are also consistent with previous knowledge on the Amazonian scale. While the earliest traces of geometric earthwork construction found in West Amazonia date back to the 1st to the 2nd millennium BCE [6, 17], most studies reported that the occupation of earthworks spread from the first millennium CE [5] to very recent periods [4, 7, 9, 18, 32], sometimes after the European Contact [4, 7], see S1 Table.

Alongside the cohort of Common Era dates, we retrieved four surprisingly old dates between the 8th and the 3rd millennium BCE (7739 to 2585 BCE, i.e. approximately 9.7 to 4.5 ka B.P.) at the sites of Nouragues and Mont Galbao, whose interpretation is uncertain. Indeed, most charcoal and geochemical studies highlight the rarity of natural fires in Amazon rainforests, thus postulating that charcoals can be attributed to human activities [51–53]. However, Amazonia experienced a drier period during the early to mid-Holocene. Cordeiro et al. [54] proposed two periods of drought in South-East Amazonia between 11.8 ka and 4.75 ka B.P. (with a maximum between 7.6 ka and 4.75 ka B.P.), while Mayle and Power [51] suggested that the dry period extended between 8 ka and 4 ka B.P. in lowland Amazonia. In French Guiana, Charles-Dominique et al. [55] highlighted instances of paleo-fires over the periods 10000–8000 B.P. and 6000–4000 B.P., well before the earliest occupations of French Guiana that are estimated at 6200 B.P. at site 'Plateau des Mines' in western French Guiana ([56], p. 135). Although the anthropogenic origin of these ancient dates cannot be ruled out, a climatic origin remains more probable. In-depth investigations of paleovegetations and paleofires in French Guiana are necessary to determine the natural or anthropogenic origin of these ancient charcoals.

## Landscape-scale spatial anthropogenic disturbances

Pre-Columbian societies had extended impacts on forest landscapes in the three study sites, not only within ring ditch enclosures but also on the surrounding territory. The composite index developed is a powerful tool to capture multi-proxy anthropogenic disturbances in our vast and continuous study areas. As expected, the composite index had slightly higher values between 0 and 30 cm depth than in deeper soil levels, where the index values were lower and globally more homogeneous. This is consistent with a previous study by Satiro et al. [57] who shown that the differences between Amazonian Dark Earths and adjacent soils were higher in the first 30 cm based on micronutrient availability. In our study, pre-Columbian disturbances were stronger in the enclosures of the ring ditches. However, our results at three independent study sites indicate that anthropogenic disturbances were not limited to the enclosures of ring ditches. Interestingly, the composite index revealed a common trend between the three sites, decreasing from ring ditch enclosures to slopes and adjacent hills. This observation is notably supported by soil color variations and the presence of macrocharcoals. However, dark-colored soils were always thinner on the adjacent hills than on the ring ditches. At MC87, the colors 7.5 YR 3/2 and 2.5 YR 3/4 extended across the first 35 cm while these soil colors were detected over the first 20 cm on the adjacent hill. At NOUR, the color 10 YR 4/4 extended across the entire profile on the ring ditch and was detected only between the surface and 5 to 15 cm on the adjacent hill. At GALB, the color 10 YR 4/4 was found up to 30 cm in the ring ditch enclosure and detected on the surface sample (0–5) in the adjacent hill. Similarly, enrichments in microcharcoals were detected not only within ring ditch enclosures but also on several pits in ring ditch slopes and adjacent hills. Contrary to *terra-firme* areas, bottomlands contain no or scarce archaeological artifacts. Nevertheless, this lack of evidence does not preclude that these areas may have been circulation zones and access to key resources not found in *terra-firme* such as water, hydrophilic plants (e.g. *Euterpe oleracea*) and gleysols.

Our results corroborate previous studies carried out in French Guiana and Amazonia, which also support the extensive use of forest landscapes by pre-Columbian societies. For example, Bodin et al. [15] suggested that anthropogenic disturbances affected forest recolonization on the adjacent hill at the ring ditch site of Saut Pararé, located about 8 km from NOUR site. In the Brazilian state of Acre, Watling et al. [33] proposed that the joint increase in palm phytoliths and microcharcoals in soil profiles at both geoglyph sites and surroundings areas (3.5 km apart) is an expression of an overall increase in human land use. Robinson et al. [58] revealed the existence of anthropogenic brown and black soils near ditched sites in northern Bolivia. Our results are therefore in line with recent findings that interfluvial regions were subjected to pre-Columbian socio-environmental dynamics. At a regional scale, LIDAR analyses revealed that pre-Columbian societies had a complex spatial organization where political, settlement, and agricultural sites were linked by causeway networks over areas of several hundred km$^2$ [4, 7]. Through a continuous and standardized sampling design, our study introduces a complementary approach to assess fine-scale anthropogenic disturbances in the landscapes surrounding ring ditches.

## Moving beyond the ADE paradigm

The soil properties within the ring ditch enclosures did not meet anthrosol criteria and cannot be considered as ADE *sensu stricto* (i.e. *terra-preta*). According to the World Reference Base for Soil Resources [59], an anthrosol is formed by long and intensive agricultural use. It is defined as a soil with a pretic horizon thicker than 50 cm, which itself is defined as a dark horizon (Munsell color value ≤4 and a chroma ≤3) enriched in Corg, P, Ca, and Mg with charcoal and/or archaeological artifacts: shards, lithic instruments, bones [37, 38, 43, 60–63]. The soils of the ring ditch enclosures revealed enrichments in artifacts, Corg, and nutrients (Ca, Mg, K) over the first 30 to 35 centimeters only, and the darkest horizons were dark reddish brown (2.5 YR 3/4) to dark yellowish brown (10 YR 4/4) with Munsell values between 3 and 4 and a chroma of 4.

Nevertheless, soil properties revealed a significant anthropogenic influence compared to adjacent soils. The concentrations of Corg (between 11.5 and 177 g/Kg), Ca (between 0.01 and 0.78 g/Kg), and Mg (between 0.001 and 0. 085 g/Kg) were lower than those commonly found in ADEs [58, 62–64] but higher than those of adjacent soils in our study sites. Indeed, soil modification by human activities is a multidirectional and gradual process that leads to various trajectories and degrees of soil transformation. Soil properties in the present study are comparable to the properties of 'transitional' soils, also known as 'Amazonian Brown Earths' (ABEs) or formerly '*terra mulata*' [58]. For example, ABEs can exhibit colors ranging from dark brown (7.5 YR 3/2) to yellowish brown (10 YR 5/6), and concentrations of Ca (0.2 to 0.7 g/Kg) and Mg (0.04 to 0.09 g/Kg) that are comparable to the concentrations found in our study [38, 58, 62]. This is also consistent with the observations by Brancier et al. [14] who argued, based on micromorphology and the abundance of anthropogenic artifacts, that the soils in MC87 were closer to '*terra mulata*' than to '*terra preta*'. Although Costa et al. [62] proposed that '*terra mulata*' is the result of ADE degradation, Robinson et al. [58] argue that these two soil types are the result of distinct processes driven by different management of resources and space. Moreover, regional differences between the Amazon basin—where most ADEs are found—and the Guiana Shield, and between fluvial and interfluvial regions—where ADEs are rare—should be taken with caution as the pedogenesis processes may differ. The specific location of our study sites in interfluvial areas of the Guiana shield argues for a better consideration of this region in our understanding of the long-term influence of human activities on soil properties and pedogenesis in Amazonia.

## Land-use and landscape management at ring ditch sites

Both long periods of occupation, artifacts and soil properties argue in favor of perennial settlements. Indeed, soils from ring ditches contained the highest occurrence of macro- and microcharcoals, ceramic shards, and significant enrichments in chemical indicators (Corg, Ntot, Ca, Mg, K). Such patterns are frequent at pre-Columbian sites and are interpreted as evidence of early domestic occupation [4, 62, 65, 66] and can unveil ancient land use and management. For example, ceramic shards were almost exclusively detected within ring ditch enclosures and ditches, suggesting housing areas. The overall higher concentrations of N, K and Ca within the ring ditches traduce a higher soil fertility. High concentrations of nutrients at pre-Columbian sites are commonly attributed to an accumulation of organic wastes of animal and plant origin [62, 67], possibly reinforced by plant ash amendments [68]. The present study also revealed the presence of macro- and microcharcoals in the three study areas. These new results are consistent with the previous study by Brancier et al. [14] that provided evidence of burned soils within the enclosure of MC87. The frequency of fire indicators in archaeological and paleoenvironmental records is commonly interpreted as a marker of fire management in Amazonia [69, 70]. For Watling et al. [33], charcoal particles larger than 100–125 μm are more likely to result from a local or supralocal fire signal, and those larger than 250 μm correspond to *in-situ* burning. Therefore, the abundance of macrocharcoals between 2 and 5 mm found in our study thus suggest *in-situ* fire management. Differences in microcharcoal concentrations between pits along the transects however suggest that not all areas were subjected to the same fire regime. Kinematic transport models and empirical meta-analyses suggest that microcharcoals (>150μm) could be transported up to fifty kilometers in the case of natural fires [71, 72]. Intensive, large-scale human-induced fires throughout ring ditch territories should have homogenized microcharcoal signals across pits. Instead, burning was more likely a local process in small patchy areas than an extensive practice affecting vast areas. Fire management may have served different domestic (e.g. forest clearance) and/or agricultural purposes that are not necessarily mutually exclusive. For example, Arroyo-Kalin [68] suggests that microcharcoal concentrations in anthropogenic soils can be interpreted as evidence of highly localized slash-and-burn cultivation. Bodin et al. [15] point out, however, that anthracological deposits alone cannot be used to decide whether charcoal is of agricultural or domestic origin. Indeed, archaeobotanical and isotopic (d$^{13}$C) analyses are necessary to investigate the extent of forest clearance, and to deepen our understanding of pre-Columbian landscape management. At NOUR, Bodin et al. [15] suggested based on anthracological assemblages that human activities favored a secondary vegetation composed of pioneer and heliophilic taxa, thus suggesting forest openings during occupation periods. Conversely, Watling et al. [33] suggested that the landscapes remained relatively closed (i.e. 'anthropogenic forests') during the occupation based on phytolith assemblages and d$^{13}$C.

Although it is impossible to conclude about the uses of pre-Columbian landscapes at ring ditch sites, our study corroborates the hypothesis of 'domestic landscapes' [24], where human occupation had extended and long-term impacts on contemporary forest landscapes, either directly (through plant cultivation for food and/or material production) or indirectly (through landscape management). Nonetheless, the present study paves the way to in-depth investigations of both contemporary and ancient vegetation to tackle the purposes of landscape management at ring ditch sites of French Guiana.

## Conclusion

Taken together, our results support the idea that pre-Columbian societies made extensive use of their landscapes in the French Guiana hinterlands, leaving lasting traces of their activities in

the soils of ring ditch sites and their surrounding territories. Soil surveys revealed anthropogenic alteration corroborated by local enrichments in chemical indicators, darker soils, and abundant macro- and microcharcoals, whose overall intensity decreased from ring ditch enclosures to surrounding areas. Furthermore, detected anthropogenic soils do not meet the characteristics of "Amazonian Dark Earths" *sensu stricto*, thus arguing for an inclusive consideration of "Amazonian Brown Earths" as Amazonian anthrosols. Finally, further investigation of ancient vegetation through phytolith and isotope analyses is essential to deepen our understanding of the exact land uses and aims of landscape management at ring ditch sites.

## Supporting information

**S1 Fig. Proportion of samples per pit containing ceramic shards (red triangles), macro-charcoals (gray squares), and soots (black circles).**
(TIF)

**S2 Fig. Distribution of microcharcoal abundance across sites (GALB, MC87, NOUR), landscape-scale localizations (ring ditch enclosure and ditch, ring ditch plateau and slope, adjacent hill) and depth ranges.**
(TIF)

**S3 Fig. Principal component analyses (PCAs) illustrating variations in soil properties between study sites in *terra-firme* areas.**
(TIF)

**S4 Fig. Principal component analyses (PCAs) illustrating variations in soil properties between landscape-scale localizations in *terra-firme* areas within each study site.** 1: Ring ditch enclosure and ditch; 2: Ring ditch plateau and slopes; 3: Adjacent hill.
(TIF)

**S5 Fig. Distribution of averaged values of physicochemical properties according to study sites, localizations, and depth of soil samples.** Averaged values are calculated for each depth from all pits in the same localization (Ring ditch Enclosure & Ditch, Ring ditch Hilltop & Slope, Bottomland, Adjacent hill).
(TIF)

**S6 Fig. Composite index of anthropogenic disturbances at MC87.** Entire soil profile (upper pane), between 0 and 30 cm deep (middle pane), between 30 and 60 cm deep (lower pane).
(TIF)

**S7 Fig. Composite index of anthropogenic disturbances at NOUR.** Entire soil profile (upper pane), between 0 and 30 cm deep (middle pane), between 30 and 60 cm deep (lower pane).
(TIF)

**S8 Fig. Composite index of anthropogenic disturbances at GALB.** Entire soil profile (upper pane), between 0 and 30 cm deep (middle pane), between 30 and 60 cm deep (lower pane).
(TIF)

**S1 Table. Non-exhaustive literature overview of earthwork radiocarbon dates in Amazonia.**
(DOCX)

**S2 Table. Radiocarbon dates obtained from the three study sites.** The calibration was performed using the OxCal 4.4 program with the IntCal20 calibration curve.
(DOCX)

**S3 Table. Generalized linear model analyses of soil physicochemical properties.** Model 1: Soil property ~ Soil type (with soil type: ferralsols, hydromorphic gleysol). Model 2: Soil property ~ Site + Localization + Depth (with site: MC87, NOUR, GALB; location: ring ditch enclosure and ditch, ring ditch hilltops and slopes, adjacent hills). * p-value < 5%, ** p-value < 1%. (DOCX)

**S1 File. Construction of the composite index of anthropogenic disturbance.** (DOCX)

**S2 File. Inclusivity in global research.** (PDF)

## Acknowledgments

The authors are grateful to all institutions that supported this study and facilitated its implementation. We acknowledge support from the CNRS Nouragues Ecological Research Station which benefits from "Investissement d'Avenir" grants managed by Agence Nationale de la Recherche (AnaEE France ANR-11-INBS-0001; Labex CEBA ANR-10-LABX-25-01). We also acknowledge support from the Parc Amazonien de Guyane (PAG) and the Regional Archaeology Department of the Direction Générale de la Cohésion et des Populations (DGCOPOP). We are also grateful to the National Forest Office (Office National des Forêts, ONF) and to the Direction de l'Environnement, de l'Agriculture, de l'Alimentation et de la Forêt (DEAAF).

We are particularly grateful to the curator of archaeology in French Guiana Régis Issenmann, the team of the CNRS research station (Nina Marchand, Marie-Françoise Lecanu, Sophie Ménager, Florian Jeanne, Patrick Chatelet, and Elodie Schloesing), Julien Cambou and Hélène Delvaux from the PAG Natural and Cultural Heritage Department, Sebastien Sant from the PAG territorial delegation in Saül who accompanied us during the first field expedition at Mont Galbao, as well as Olivier Brunaux from the ONF for his help to trace the track to MC87. We are also grateful to Stéphane Calmant, our IRD representative in French Guiana, as well as all the administrative staff of IRD Cayenne for their kind welcome and support. We are finally grateful to Cyril Gaertner and Michèle Pernak (CNRS, UAR LEEISA) for their assistance in the field at Nouragues.

## Author Contributions

**Conceptualization:** Marc Testé, Julien Engel, Mickael Mestre, Louise Brousseau.

**Data curation:** Marc Testé, Julien Engel, Louise Brousseau.

**Formal analysis:** Marc Testé, Louise Brousseau.

**Funding acquisition:** Louise Brousseau.

**Investigation:** Marc Testé, Julien Engel, Kevin Mabobet, Louise Brousseau.

**Methodology:** Marc Testé, Julien Engel, Louise Brousseau.

**Project administration:** Louise Brousseau.

**Supervision:** Louise Brousseau.

**Validation:** Louise Brousseau.

**Visualization:** Marc Testé, Julien Engel, Louise Brousseau.

**Writing – original draft:** Marc Testé, Julien Engel, Mickael Mestre, Louise Brousseau.

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
