## [Decision Letter · Decision Letter 0]

1 Apr 2024

PONE-D-24-03891Landscape-scale spatial variations of pre-Columbian anthropogenic disturbances at three ring ditch sites in French GuianaPLOS ONE

Dear Dr. TESTÉ,

Thank you for submitting your manuscript to PLOS ONE. After careful consideration, we feel that it has merit but does not fully meet PLOS ONE’s publication criteria as it currently stands. Therefore, we invite you to submit a revised version of the manuscript that addresses the points raised during the review process.

We look forward to receiving your revised manuscript.

Kind regards,

John P. Hart, Ph.D.

Academic Editor

PLOS ONE

Journal Requirements:

"DOPAMICS is funded by the European Union under grant no. 101039272. Views and opinions expressed are those of the authors only and do not necessarily reflect those of the European Union or the European Research Council Executive Agency (ERCEA). Neither the European Union nor the granting authority can be held responsible for them."

6. We notice that your supplementary tables are included in the manuscript file. Please remove them and upload them with the file type 'Supporting Information'. Please ensure that each Supporting Information file has a legend listed in the manuscript after the references list.

**Additional Editor Comments:**

I have been able to obtain only one review. However, the review is very comprehensive so I feel comfortable making a decision based on this one review. Please address all of the reviewer's comments and suggestions while making your revisions.

Reviewers' comments:

Reviewer's Responses to Questions

**Comments to the Author**

1. Is the manuscript technically sound, and do the data support the conclusions?

Reviewer #1: Yes

2. Has the statistical analysis been performed appropriately and rigorously? 

Reviewer #1: I Don't Know

3. Have the authors made all data underlying the findings in their manuscript fully available?

Reviewer #1: Yes

4. Is the manuscript presented in an intelligible fashion and written in standard English?

Reviewer #1: Yes

5. Review Comments to the Author

Reviewer #1: Comments on the paper :

PONE-D-24-03891

Landscape-scale spatial variations of pre-Columbian anthropogenic disturbances at three ring ditch sites in French Guiana

This study investigates past anthropogenic disturbances in French Guiana by analysing several indicators from the soil. The authors studied three pre-Columbian sites presenting circular ditches located on top of hills, and took into account the slopes of the hills, the bottomlands and adjacent hills, following transects, to extend the investigation of past disturbances not only on the ditch sites, but also on their surrounding areas. They analysed the charcoal concentration, the soil texture and colour and the presence of archaeological artefacts to estimate the level of anthropogenization of the soil in these different areas. They also performed radiocarbon dating to assess the period of occupation on the sites of interest.

I find this study very interesting as it is one of the very few to address pre-Columbian impacts on the landscape in the Guiana Shield. It contributes to the knowledge of Anthropogenic Dark Earths, showing that the dichotomy that is often done for Amazonia (terra mulata vs terra preta) is a bit too simplistic and that it exists a broad range of Anthropogenic Soils. I like the disturbance index that the authors introduce in this study: it is very practical given the number of proxies that can be taken into account when we talk about anthropogenic markers in soils. Taking into account the surrounding landscape and not only the ditch sites is also interesting, as we can see that there is a sort of gradient in the anthropogenic markers in the soil. The discussion is interesting and answers the questions presented at the end of the introduction. But maybe it could go a bit further, by considering a bit more archaeological literature when it comes about the uses of the sites.

Below are my remarks, questions and comments.

In the introduction, I noted typos abouts Llanos de Mojos, written differently sometimes.

Line 160: what does NT0125 stand for?

Figure 1: Could you add the sites on the altitude diagram? I think you should write ‘Parc Amazonien de Guyane’ instead of PAG here, because we see this Fig. before you give the meaning of the acronym line 191.

Line 165: here Mont-Galbao but later Mont Galbao without hyphen.

Line 168: maybe you could specify what MC stands for, those who do not know will wonder.

Line 179: lowercase here for the coordinates x and y, but uppercase line 168 (X, Y).

Line 191: maybe just put PAG between brackets?

Line 203: ‘with little or no evidence of pre-Columbian disturbance’. In terms of what? Regarding the floristic composition/structure? No potsherds? No charcoal?

Line 206: I am not sure I got it right: you got samples every 5 cm, does that mean every sample is 5 cm-thick, from the surface to 5 cm, then from 5 to 10 cm and so on? I think you should make it clearer as you did lines 224-227.

Line 214: I was already wondering when reading the previous paragraph, what is the volume of each soil sample?

Line 261: when charcoal is anatomically identifiable, it is better to do so before doing radiocarbon dating. You might have dated several times the same tree/shrub, hence getting the same age. It could be the case in your data, as some dates are fairly similar. It is especially likely when charcoal pieces come from samples a few cm apart. Maybe it is not a big deal for the aim of your study, but it still may be a bias regarding the information on pre-Columbian occupation.

Figure 2: I think you should specify what the red and green rectangles mean in the legend.

Lines 310-311: when you say ‘global proportion’, do you mean the ‘mean proportion’? (same question for the other paragraphs).

I am a bit confused that you emphasize the peaks on the main hill, where the enclosure is, but not the peak on the adjacent hill. What I mean is that it already puts some direction, some point of view, in your results, but I would expect that later in the discussion (if relevant). In the same way, you specify that C7 contained neither soot nor charcoal, but it is also true for C12. At this point, the reader does not know if the adjacent hill had a meaning, a function (despite the absence of a ditch), so one may wonder why you treat it differently.

Lines 329-331: I think it reinforces my comment above.

Lines 375-381: I am not convinced that it makes sense to include the adjacent hills when you compare the sites. They are different entities from the ring ditch sites, maybe they had a completely different function. They probably create outliers. What if you remove them? Would the sites be still different regarding the microcharcoal concentration? I think you should only consider the hilltop and the slopes of a same hill as a site (for this analysis) (maybe I am wrong but then I think you should explain your choice). About this paragraph, I think it should come after the description of microcharcoal content in MC87, NOUR and GALB.

I think you should be consistent with the order of the paragraphs in the results and in the M&M. If you first introduce microcharcoal in the M&M, before physico-chemistry, I think you should do the same in the results.

About these grey bands in Supp. Fig. 2: what are they for? They hide information.

Line 387-388: what do you mean ‘consistent with the landscape-scale position of pits’? I find it unclear.

Fig. 5: I find it a bit confusing that the sites do not appear here, as you said in the M&M section that you performed the PCAs to visualize variations in soil properties across study sites too. I think it would be very interesting if you could do it, especially when you show that there are significant differences in chemical elements between sites.

Supp. Fig. 3: What do the brown stripes in the vertical rectangles on the left mean? Do they represent the depth? Why are some stripes thicker than others? Why is there no information on grain size for the upperpart (I guess) of the bottomland sample at GALB?

Suppl. Method: I see rectangles where I guess some special characters or formulas should appear (line 852), I guess something happened during the uploading of the manuscript.

Suppl. Fig. 4 to 6: I find these figures very interesting and I think they should go in the main text, maybe even in place of Fig. 3 and 4 because they sum up their information. Just a comment for visualization purposes: I think it would be better to represent the two panes of a same figure in the same way. I mean, on the left pane the adjacent hills are at the bottom, but they are on the top on the right pane.

Line 449: I would not say ‘comprehensive’, because it is not. It sure uses a multiproxy approach, but many other proxies could have been used. Please get me right: I just think it is not the best term to employ here, I do not criticize the number of proxies you have used.

Discussion

Line 465: the name is missing before the citation number 15.

Line 475: there is no anthracological studies in the references you cite, they are not dealing with charcoal anatomical identification ; ‘charcoal studies’ would be better.

Line 489: I think you should add a couple references here, at the end of the first sentence of the paragraph.

Line 501: ‘only between the surface and 5 to 15 cm’ on the adjacent hill?

Lines 488-507: there are no references in this paragraph but this trend (anthropogenic disturbance indicators in the first 30 cm or so) has been observed in other studies (I am sorry I do not have them in mind so maybe my comment is useless but I am pretty sure of that).

Line 556-557: I agree that most ceramics are certainly found in enclosures but they also can be found elsewhere, sometimes quite abundantly, in uprooting mounds for example.

Line 565: about charcoal particle size; I think you can strengthen your discussion here by adding a bit more references about charcoal particle motion, transport and size. I am thinking about the work by Clark (Clark, J. S. (1988). Particle motion and the theory of charcoal analysis: source area, transport, deposition, and sampling. Quaternary research, 30(1), 67-80.), to cite only one.

Line 576: I do not think archaeobotanical and isotopic analyses alone can make clear the exact uses of managed landscapes, because it can be extremely tricky to interpret these data without the insights from other fields (archaeology, soil science for example). So I think you should not use the terms ‘unveil the exact uses’.

Line 584: Careful with the term agriculture or derivates, better use food production (see for example Piperno, D. R. (2011). The origins of plant cultivation and domestication in the New World tropics: patterns, process, and new developments. Current anthropology, 52(S4), S453-S470.)

Line 586: in the line of my comment above, I would not say ‘exact’ (but maybe I am too pessimistic!)

Line 596: why phytoliths especially? Phytoliths would help for sure, but so would pollen or isotopes or many other proxies or fields of research!

Line 608: There is a typo: Régis Issenmann.

6. PLOS authors have the option to publish the peer review history of their article (what does this mean?). If published, this will include your full peer review and any attached files.

Reviewer #1: No

---

## [Author Response · Author response to Decision Letter 0]

21 May 2024

Dear reviewer,

We thank you very much for your useful review of our manuscript. We took into account your comments and improved the manuscript accordingly. 

Please find our answers below. 

Kind regards,

Marc Testé

Line 160: what does NT0125 stand for?

NT0125 corresponds to the scientific code established by the WWF to designate the "Guianan Moist Forests" ecoregion (https://www.worldwildlife.org/ecoregions/nt0125). The listing of ecoregions now managed by One Earth (https://www.oneearth.org/ecoregions/guianan-lowland-moist-forests/). We thus replaced NT0125 by the nomenclature of the bioregion (NT21) and ecoregion (465), and added the corresponding link in the manuscript. 

Figure 1: Could you add the sites on the altitude diagram? I think you should write ‘Parc Amazonien de Guyane’ instead of PAG here, because we see this Fig. before you give the meaning of the acronym line 191.

We have added the location of the sites on the altitude diagram and explained the PAG acronym in English/French.

Line 165: here Mont-Galbao but later Mont Galbao without hyphen.

We homogenized ''Mont Galbao'' without the hyphen.

Line 168: maybe you could specify what MC stands for, those who do not know will wonder.

We specified at lined 64 that ring ditches are locally named "Montagne Couronnée" and abbreviated "MC".

Line 179: lowercase here for the coordinates x and y, but uppercase line 168 (X, Y).

We have detected a mistake in the GPS coordinates and updated the manuscript accordingly. We formatted the coordinates as decimal degree (WGS84) and specified “N” and “E” where applicable.

Line 191: maybe just put PAG between brackets?

Done

Line 203: ‘with little or no evidence of pre-Columbian disturbance’. In terms of what? Regarding the floristic composition/structure? No potsherds? No charcoal?

We modified the text accordingly: : “i.e., no earthwork, absence of or few shards and charcoals in treefall mounds”.

Line 206: I am not sure I got it right: you got samples every 5 cm, does that mean every sample is 5 cm-thick, from the surface to 5 cm, then from 5 to 10 cm and so on? I think you should make it clearer as you did lines 224-227.

We modified the text accordingly: “For each pit, 5 cm-thick soil samples were collected from the surface (0-5cm) to a depth of 120 cm (115-120cm), i.e. a total of 24 samples per pit.”

Line 214: I was already wondering when reading the previous paragraph, what is the volume of each soil sample?

We did not measure the raw volume of the samples in the field but rather weighted the dry mass in the lab (after oven drying). The dry mass of each sample is provided in the data. Because the soil volume per sample was low (below 100mL/ sample) and the dry mass heterogeneous across samples after drying, we decided to describe artifacts as presence/absence only and not as a quantitative variable (except for microcharcoals for which we sub-sampled 2.5 cm3 as described in the corresponding section “Microcharcoal Abundance”).

Line 261: when charcoal is anatomically identifiable, it is better to do so before doing radiocarbon dating. You might have dated several times the same tree/shrub, hence getting the same age. It could be the case in your data, as some dates are fairly similar. It is especially likely when charcoal pieces come from samples a few cm apart. Maybe it is not a big deal for the aim of your study, but it still may be a bias regarding the information on pre-Columbian occupation.

We have not identified the macrocharcoals due to a lack of anthracological expertise in our team. We dated a cohort of 10 charcoals per site from different pits and depth among available charcoals to limit this bias and to get a large range of dates. The consistency of dates obtained between sites made us confident about the global ranges of occupation at ring ditch sites in French Guiana. 

However, we agree with your comment and mentioned this potential bias in the corresponding section of the discussion as follow: “Although we did not carry out a taxonomic identification of the macrocharcoals selected for radiocarbon dating, we dated a cohort of ten charcoals per site from different pits and depth to get a large range of dates and avoid dating the same plant. The consistency of dates obtained between the three study sites made us confident about the periods of occupation.”. 

Figure 2: I think you should specify what the red and green rectangles mean in the legend.

Done

Lines 310-311: when you say ‘global proportion’, do you mean the ‘mean proportion’? (same question for the other paragraphs). 

We mean the total proportion, i.e., the total number of samples containing macrocharcoals divided by the total number of samples in the different zones (all pits confounded). We replaced “global” by “total” in all paragraphs and removed the term “pits” to make it clearer that it is not a mean across pits.

I am a bit confused that you emphasize the peaks on the main hill, where the enclosure is, but not the peak on the adjacent hill. What I mean is that it already puts some direction, some point of view, in your results, but I would expect that later in the discussion (if relevant). 

At MC87, the pits containing the highest proportion of macrocharcoals and soots (within pits) were detected on the ring ditch enclosure (66.67% for macrocharcoals, 50% for soots). We added a sentence describing the proportion of macrocharcoals and soots on the adjacent hill to make it clearer: “Macrocharcoal peaks were also identified on the slope of the adjacent hill at C14 and C15 (41.67 % of the samples containing macrocharcoals), while the proportion of soots was equal or below 25% (C16).”.

In the same way, you specify that C7 contained neither soot nor charcoal, but it is also true for C12. At this point, the reader does not know if the adjacent hill had a meaning, a function (despite the absence of a ditch), so one may wonder why you treat it differently.

We modified the last sentence and listed all pits without macrocharcoals and soots: “Several pits contained neither macrocharcoals nor soots: C7 (bottom of the slope of the ring ditch), C8 to C12 (bottomland), AH1, AH2 and C19 (adjacent hilltop)”.

Lines 329-331: I think it reinforces my comment above.

We added a sentence in the paragraph: “Secondary peaks of macrocharcoals (33.3%) and soots (41.6%) were observed in pit C23 that is located on the adjacent hill.” 

Lines 375-381: I am not convinced that it makes sense to include the adjacent hills when you compare the sites. They are different entities from the ring ditch sites, maybe they had a completely different function. They probably create outliers. What if you remove them? Would the sites be still different regarding the microcharcoal concentration? I think you should only consider the hilltop and the slopes of a same hill as a site (for this analysis) (maybe I am wrong but then I think you should explain your choice). 

We carried out the analysis suggested (Microcharcoal count ~ Site + Depth after removal of adjacent hills) and the site effect remained significant (p-value = 3.47 x 10-11). We added the results of this complementary analysis in the Results as follow : “Site effect remained significant when considering microcharcoal counts of the ring ditches only, i.e. after removal of the adjacent hills and testing Site and Depth effects only: Microcharcoal count ~ Site + Depth (p-value = 3.47 × 10-11).” 

About this paragraph, I think it should come after the description of microcharcoal content in MC87, NOUR and GALB.I think you should be consistent with the order of the paragraphs in the results and in the M&M. If you first introduce microcharcoal in the M&M, before physico-chemistry, I think you should do the same in the results.

We modified the order of paragraphs in M&M. 

About these grey bands in Supp. Fig. 2: what are they for? They hide information.

We added grey bands in Supp. Fig. 2 to help the reader but the bands overlapped with the plot while converting it in TIFF format. We corrected it. 

Line 387-388: what do you mean ‘consistent with the landscape-scale position of pits’? I find it unclear.

We explained it in the corresponding sentence as follow “i.e., highest peaks of microcharcoals were found within (C15, C13) or close to (C10, C7) the ring ditch enclosure”. 

Fig. 5: I find it a bit confusing that the sites do not appear here, as you said in the M&M section that you performed the PCAs to visualize variations in soil properties across study sites too. I think it would be very interesting if you could do it, especially when you show that there are significant differences in chemical elements between sites.

We added the PCA with different colors for the different sites in supplementary Figure 3. Please note that it is the same PCA as in figure 5 lower pane ; only the colors of groups has changed. We did not included it in the main text as we wanted to illustrate the difference between landscape-scale localization rather than the differences between study sites. We also added a supplementary figure 4 with PCAs showing differences between localizations within each study site.

Supp. Fig. 3: What do the brown stripes in the vertical rectangles on the left mean? Do they represent the depth? 

Yes, the brown stripes correspond to the depths of the samples selected for physico-chemistry. We specified it in the figure. 

Why are some stripes thicker than others? 

In bottomland, dry soil samples were always below 100g due to important water-logging. To remedy this, we pooled surface samples up to 20 cm as explained in the section “Physico-chemistry” in M&M. 

Why is there no information on grain size for the upperpart (I guess) of the bottomland sample at GALB?

For GALB, the dry mass of surface samples (above 30cm) were unfortunately insufficient for particle size analysis. We specified it in the figure legend. 

Suppl. Method: I see rectangles where I guess some special characters or formulas should appear (line 852), I guess something happened during the uploading of the manuscript.

It's a problem of format at the time of conversion to pdf.

The conversion into pdf has removed the "Ppit,d" characters during the submission process. We do apologize and will check that the formatting is correct when re-submitting/proofreading. 

Suppl. Fig. 4 to 6: I find these figures very interesting and I think they should go in the main text, maybe even in place of Fig. 3 and 4 because they sum up their information. Just a comment for visualization purposes: I think it would be better to represent the two panes of a same figure in the same way. I mean, on the left pane the adjacent hills are at the bottom, but they are on the top on the right pane.

We think it is a pity to remove the figures 3 and 4. We added a figure 6 in the main text providing the composite index between 0 and 30 cm and let the full figures supl. Figures 6 to 8. We also modified the order of boxes in the left pane as suggested. 

Line 449: I would not say ‘comprehensive’, because it is not. It sure uses a multiproxy approach, but many other proxies could have been used. Please get me right: I just think it is not the best term to employ here, I do not criticize the number of proxies you have used.

We do agree and replaced “comprehensive” by “extensive”.

Line 465: the name is missing before the citation number 15

Done

Line 475: there is no anthracological studies in the references you cite, they are not dealing with charcoal anatomical identification ; ‘charcoal studies’ would be better.

We replaced “anthracological” by “charcoal” studies. 

Line 489: I think you should add a couple references here, at the end of the first sentence of the paragraph.

This topic sentence summarizes our finding on the three ring ditch sites under study. We specified it in the sentence. 

Line 501: ‘only between the surface and 5 to 15 cm’ on the adjacent hill?

We do apologize. part of the sentence was missing. We completed it by adding “on the adjacent hill”.

Lines 488-507: there are no references in this paragraph but this trend (anthropogenic disturbance indicators in the first 30 cm or so) has been observed in other studies (I am sorry I do not have them in mind so maybe my comment is useless but I am pretty sure of that).

We added a reference and a sentence in the paragraph: “This is consistent with a previous study by Satiro et al. who shown that the differences between Amazonian Dark Earths and adjacent soils were higher in the first 30 cm based on their physico-chemical properties”. 

Line 556-557: I agree that most ceramics are certainly found in enclosures but they also can be found elsewhere, sometimes quite abundantly, in uprooting mounds for example.

We did not quantified shard abundance in our samples nor in uprooting mounds. This would imply to have sampled and sieved important volumes of soils (e.g. 5L/sample) which was not possible in our protocol that involved many sampling points and depths (totaling about 1000 samples).

Line 565: about charcoal particle size; I think you can strengthen your discussion here by adding a bit more references about charcoal particle motion, transport and size. I am thinking about the work by Clark (Clark, J. S. (1988). Particle motion and the theory of charcoal analysis: source area, transport, deposition, and sampling. Quaternary research, 30(1), 67-80.), to cite only one.

We added a sentence and corresponding references in the paragraph (recent kinematic transport models by Vachula et al.) to illustrate our purpose. 

Line 576: I do not think archaeobotanical and isotopic analyses alone can make clear the exact uses of managed landscapes, because it can be extremely tricky to interpret these data without the insights from other fields (archaeology, soil science for example). So I think you should not use the terms ‘unveil the exact uses’.

We modified the sentence accordingly: “[…] to deepen our understanding of pre-Columbian landscape management.

Line 584: Careful with the term agriculture or derivates, better use food production (see for example Piperno, D. R. (2011). The origins of plant cultivation and domestication in the New World tropics: patterns, process, and new developments. Current anthropology, 52(S4), S453-S470.)

Plant cultivation by traditional societies is not limited to food production, as traditional ecological knowledge highlight multiple use of plants in the Amazon (e.g., basketry, medicine). In her article, Dolores Piperno mentions the term “plant cultivation” (“It is becoming clear that the more interesting question may be the origins of plant cultivation rather than the origins of agriculture.”). We removed “by early culturist” and completed our sentence by “for food and/or material production”

Line 586: in the line of my comment above, I would not say ‘exact’ (but maybe I am too pessimistic!)

We've removed the word "exact".

Line 596: why phytoliths especially? Phytoliths would help for sure, but so would pollen or isotopes or many other proxies or fields of research!

 Phytoliths are bioindicators that can give ecological and taxonomic information on local palaeovegetations. Of course, pollen and isotopes can also bring very useful and complementary information, although ancient pollen records are found in wetlands but and are not preserved in terra-firme soils. We thus mentioned isotopes in the sentence too.

Line 608: There is a typo: Régis Issenmann.

Done

---

## [Decision Letter · Decision Letter 1]

2 Jul 2024

Landscape-scale spatial variations of pre-Columbian anthropogenic disturbances at three ring ditch sites in French Guiana

PONE-D-24-03891R1

Dear Dr. TESTÉ,

We’re pleased to inform you that your manuscript has been judged scientifically suitable for publication and will be formally accepted for publication once it meets all outstanding technical requirements.

Kind regards,

Christian Reepmeyer, PhD

Academic Editor

PLOS ONE

Additional Editor Comments (optional):

Reviewers' comments:

Reviewer's Responses to Questions

**Comments to the Author**

1. If the authors have adequately addressed your comments raised in a previous round of review and you feel that this manuscript is now acceptable for publication, you may indicate that here to bypass the “Comments to the Author” section, enter your conflict of interest statement in the “Confidential to Editor” section, and submit your "Accept" recommendation.

Reviewer #1: All comments have been addressed

2. Is the manuscript technically sound, and do the data support the conclusions?

Reviewer #1: Yes

3. Has the statistical analysis been performed appropriately and rigorously? 

Reviewer #1: Yes

4. Have the authors made all data underlying the findings in their manuscript fully available?

Reviewer #1: Yes

5. Is the manuscript presented in an intelligible fashion and written in standard English?

Reviewer #1: Yes

6. Review Comments to the Author

Reviewer #1: Dear authors,

I apologize for the delay, due to sickness.

I am very satisfied with your corrections and answers and think that your article is ready for publication.

7. PLOS authors have the option to publish the peer review history of their article (what does this mean?). If published, this will include your full peer review and any attached files.

Reviewer #1: No

---

## [Editor Report · Acceptance letter]

1 Aug 2024

PONE-D-24-03891R1 

PLOS ONE

Dear Dr. TESTÉ, 

I'm pleased to inform you that your manuscript has been deemed suitable for publication in PLOS ONE. Congratulations! Your manuscript is now being handed over to our production team.

Kind regards, 

on behalf of

Dr. Christian Reepmeyer 

Academic Editor

PLOS ONE